# Microplastics in Ecosystems: From Current Trends to Bio-Based Removal Strategies

**DOI:** 10.3390/molecules25173954

**Published:** 2020-08-30

**Authors:** Solange Magalhães, Luís Alves, Bruno Medronho, Anabela Romano, Maria da Graça Rasteiro

**Affiliations:** 1CIEPQPF, Department of Chemical Engineering, University of Coimbra, Pólo II – R. Silvio Lima, 3030-790 Coimbra, Portugal; luisalves@ci.uc.pt; 2MED–Mediterranean Institute for Agriculture, Environment and Development, Campus de Gambelas, Faculty of Sciences and Technology, University of Algarve, Ed. 8, 8005-139 Faro, Portugal; bfmedronho@ualg.pt (B.M.); aromano@ualg.pt (A.R.); 3Fibre Science and Communication Network (FSCN), Mid Sweden University, SE-851 70 Sundsvall, Sweden

**Keywords:** microplastics, lignocelluloses, flocculants, ecosystems, wastewater, removal

## Abstract

Plastics are widely used due to their excellent properties, inexpensiveness and versatility leading to an exponential consumption growth during the last decades. However, most plastic does not biodegrade in any meaningful sense; it can exist for hundreds of years. Only a small percentage of plastic waste is recycled, the rest being dumped in landfills, incinerated or simply not collected. Waste-water treatment plants can only minimize the problem by trapping plastic particles of larger size and some smaller ones remain within oxidation ponds or sewage sludge, but a large amount of microplastics still contaminate water streams and marine systems. Thus, it is clear that in order to tackle this potential ecological disaster, new strategies are necessary. This review aims at briefly introducing the microplastics threat and critically discusses emerging technologies, which are capable to efficiently clean aqueous media. Special focus is given to novel greener approaches based on lignocellulose flocculants and other biomaterials. In the final part of the present review, it was given a proof of concept, using a bioflocculant to remove micronized plastic from aqueous medium. The obtained results demonstrate the huge potential of these biopolymers to clean waters from the microplastics threat, using flocculants with appropriate structure.

## 1. Background: Introducing Plastics and Microplastics

Since the production of the first plastic material in 1907, it has been shown that these products are of great use since they are lightweight, durable, inert, and corrosion-resistant products [1,2].

Nowadays, the most commonly used synthetic polymers are: polyethylene terephthalate (PET), polyethylene of various densities (PE: LDPE, LLDPE, HDPE), polypropylene (PP), polystyrene (PS), polyvinyl chloride (PVC), acrylonitrile butadiene styrene (ABS), polylactic acid (PLA), polyamides (PA), polycarbonate (PC), and polyurethanes (PU) [2,3,4] (Figure 1).

Plastics are virtually present in all industrial sectors and their favourable properties have triggered their massive production/use. For example, 1 million plastic bottles are bought every minute around the world, that number being expected to top half a trillion by 2021 [5]. Less than half of those bottles end up getting recycled. Nearly 2 million single-use plastic bags are distributed worldwide every minute [6]. Half a billion plastic straws are used every day in USA alone, which is enough to circle the Earth three times. On the other hand, polyester and polyamide are widely used in the textile sector, in particular synthetic clothes, while polyethylene is used in personal care products as well as food packaging films [4,7].

Intimately related, microplastics are defined as plastic particles with less than 5 mm in length that are either formed by the fragmentation of larger plastics or are intentionally produced and added, for instance, into cosmetics and other personal care products [2,8,9].

Microplastics can be generally divided in two big families: primary microplastics and secondary microplastics [2,10,11,12]. Plastics with microscopic size, ca. 2–5 mm in diameter, are defined as primary microplastics [13,14,15]. Secondary microplastics enclose tiny plastic fragments derived from the breakdown of larger plastic debris, both at sea and on land. Over time, a combination of physical, biological and chemical processes can reduce the structural integrity of plastic debris, resulting in their fragmentation [16,17]. As can be seen in Figure 2, this comprises a very heterogeneous assemblage of pieces that vary in size, shape, colour, specific density, chemical composition and other characteristics which, as it will be discussed later, should be considered during the development of appropriate methods for their extraction and characterization [18,19].

The formation of secondary microplastics, which originate by the slow deterioration of macroplastics in water, comprises three different mechanisms: (1) biodeterioration, which is the modification of mechanical, chemical, and physical properties of the polymer due to the growth of microorganisms on or inside the surface of the polymers; (2) biofragmentation, which is the conversion of polymers to oligomers and monomers by the action of microorganisms; and (3) assimilation, where microorganisms are supplied with the necessary carbon, energy and nutrient sources from the fragmentation of polymers and convert carbon of plastic to CO_2_, water and biomass [20]. Yet, microplastics can also suffer photo-degradation, when exposed to prolonged periods of sunlight, because ultraviolet radiation causes oxidation of the polymer matrix, leading to bond cleavage [21]. As mechanical integrity of plastics invariably depends on their molecular-weight, any degradation of significant extent will inevitably weaken the material [22]. With the loss of structural integrity, these plastics are increasingly susceptible to fragmentation resulting from abrasion, wave-action, and turbulence [6]. This has been argued to be a continuous process, where microplastics might actually further degrade into nanoscale dimensions (so called nanoplastics). At present, the smallest particles reported in the oceans have 1.6 μm in diameter [23,24,25]. The concentration of microplastics in the surface layer of the oceans is lower than expected, mainly due to aggregation processes driven by biofilm covering of microplastics with marine biogenic particles, and subsequent sedimentation [26].

## 2. Plastics, Microplastics and Their Impact in the Ecosystems

Despite the extensive use of plastics, microplastics have recently become a central discussion theme around the world, not because of their societal benefits but mainly due to their potential nefarious impact on different ecosystems. Increasing evidences tend to show that microplastics have a negative impact, particularly on the marine environment and biota, leading to increasing environmental awareness [1,27].

Plastic pollution (and, consequently, the amount of microplastics) in the oceans is significantly growing. Normally, high-density particles sink and accumulate in the sediment, while low-density particles float at the sea surface. However, due to their small dimensions, microplastics are easily available for uptake to a wide range of marine organisms [27,28]. This reduction in size not only impairs the cleanliness of the waters, but also causes some marine animals death due to ingestion. Due to their hydrophobic properties, microplastics can also adsorb and enter the marine food-webs [23,24]. In Figure 3 and Figure 4, examples of real effects of plastic materials in marine organisms are illustrated. A comprehensive report presented by Murray identifies some impacted species, the materials involved, as well as location and source [23]. The identified species include turtles; penguins; albatrosses; petrels and shearwaters; shorebirds; skuas; gulls and auks; coastal birds other than seabirds such as baleen whales, toothed whales and dolphins; earless or true seals; sea lions and fur seals; manatees and dugong; sea otters; fish; and crustaceans [23]. The report from the “Task Group 10” from the European Commission about marine litter has reached similar conclusions [23,29,30].

Many properties that humankind find so desirable in plastic materials are also the same responsible for the plethora of problems they are globally creating, particularly in marine environments. Recently, Barboza et al. (2020) reported that any marine organisms, such as crustaceans, molluscs and fish when in exposure to microplastics may induce physical and chemical toxicity, including genotoxicity, oxidative stress, changes in behaviour, reproductive impairment, mortality, population growth rate decrease, transgenerational effects, among several others [31]. The authors also estimate that microplastics are potentially taken in by human adults from fish consumption. Another study presented by de Sá, et al. (2018) demonstrated the different ecotoxicological effects per polymer/plastic family (Figure 5). Studies were defined according to the type of MPs, groups of organisms and effects. It is clear the potential effects of MPs on aquatic biota and the problems caused every day in a large number of organisms when exposed to these particles. This exposure may cause a variety of effects and threaten individuals of many different species, in the ecosystems they live in and, ultimately, humans [32].

Although plastics are generically regarded as inert materials, with low/negligible chemical reactivity [33], there has been questions about whether its dissemination in the body and organs may not be the cause of different health issues [34]. Additionally, depending on the source (e.g., ship hull paints, fibres, rubber, polyethylenes, polypropylenes, or others), plastic may also contain different types of chemical additives which, due to their nature, may also be absorbed, contributing to unexpected adverse health problems [35]. For instance, it has been observed that the absorption of microplastics and nanoplastics by humans can lead to a wide range of organism’s obstruction, inflammation and accumulation in organs after translocation [36]. As stated by Pete Myers, founder and chief scientist of the nonprofit Environmental Health Sciences, the ingestion of microplastics can expose humans to different chemicals found in their compositions known to induce harmful effect and connected to different health issues, such as reproductive harm and obesity, organ problems and developmental delays in children. These microplastics could potentially leach their constituents, adsorbed contaminants and pathogenic organisms. On the other hand, reduction in photosynthesis of plants and effects on the feeding activity of zooplankton and marine animals (adverse effects to gill, stomach and alterations in histology) have also been argued to occur after microplastic intake [29,30]. These and other effects are well described and summarised in the recent review by Prata et al. [12]. In a recent SAPEA report [37], although the authors recognize limitations in the measurement methods currently available, they state that the concentration levels measured in many real-world locations are well below the concentration threshold observed to induce harmful effect on living creatures. Moreover, the potential nanoplastics effects are even more obscure due to the problems associated to their identification and evaluation. It is important to note that microplastic toxicity greatly depends on its concentration but other parameters, such as particle features, adsorbed contaminant agents, tissues involved, individual susceptibility, etc, may play an important role in their final toxic effect and this still remains to be properly addressed and clarified by the scientific community. Due to their capacity to move between different tissues of plants and animals, along the food chain, and strong indications of associated health issues, their “inert” features must be seriously reconsidered. It is clear that the studies regarding the microplastic toxicity, particularly to humans, are still very scarce, and thus, it definitely requires future research. The fact that there are no population studies on human health effects, has been motive of concern by a group of chief scientific advisors to the European Commission [38]. This independent expert group has recommended the EU to not only identify the most polluting activities with current or new legal actions but also establish a global scientific platform to promote microplastic research. These actions should be politically and socio-economically feasible and hopefully contribute to clarify the potential threat of microplastics and guide the future action policies of EU.

Plastic litter with a terrestrial source contributes to 80% of the plastics found in marine litter. Such plastics include primary microplastics used in cosmetics and air-blasting, improperly disposed ‘‘user’’ plastics and plastic leachates from refuse sites [1]. With approximately half the world’s population living within 100 kilometres of the coast, these types of plastic are highly prone to enter the marine environment via rivers and wastewater systems, or by being blown off-shore [6]. These microplastics can also enter waterways via domestic or industrial drainage systems [39]; whilst wastewater treatment plants trap macroplastics and some small plastic debris within oxidation ponds or sewage sludge, a large amount of microplastics passes through the filtration systems [40,41]. Plastics that enter river systems, either directly or within wastewater effluent or in refuse site leachates, will then be transported out to sea. Numerous studies have clearly shown how the high unidirectional flow of freshwater systems drives the movement of plastic debris into the oceans [1,42,43].

Besides the migration of microplastics from terrestrial sources, changes in marine equipments, such as transportation and fishing ships, contribute to the increase of plastic litter on the sea. For example, prior to the 1950s, all rope and cordage used in marine activities, including fisheries, were made of natural fibres-typically Indian or Manila hemp and cotton. These systems were often strengthened with a coating of tar or strips of worn canvas. Once these materials lost their resilience in continuous usage or if they were lost /discarded at sea, they tended to quickly degrade. However, for durability and performance reasons, these natural fibres have been replaced by nylon and other synthetic materials that are generally buoyant and far more endurable [1].

In order to face the plastic problem, research on bioplastics has been growing enormously. Environmentally friendly alternatives to traditional plastics are synthesized from biomass and renewable resources, such as Poly(lactic acid) (PLA) and Polyhydroxyalkanoate (PHA), or from fossil fuel including aliphatic plastics, such as Polybutylene succinate (PBS), which can also be used as a substrate to microorganisms [20]. With the production of bioplastics, it is also possible to reduce the CO_2_ emission, which helps minimize the greenhouse effect. Thus, the development of biodegradable plastics is often seen as a viable alternative for traditional plastics. However, they can also be a source of microplastics [44]. Biodegradable plastic materials are typically composed of synthetic polymers combined with biodegradable vegetable polymers, such as starch or vegetable oils, or with specialized chemicals (e.g., TDPA™) designed to enhance the kinetics of degradation [39]. If properly disposed, these systems are expected to decompose in industrial composting plants using appropriate heat, humidity and ventilation conditions. However, this decomposition is only partial; while components such as starch decompose, the synthetic polymers will be left behind untouched [21]. Moreover, in the relatively cold marine environment and in the absence of terrestrial microbes, decomposition rates of even degradable bioplastic components will be significantly prolonged, increasing the likelihood of plastic fouling and, subsequently, reducing the UV permeation, essential for the degradation process [45]. At the end, when decomposition finally occurs, microplastics are released into the marine environment, contributing to sea water pollution. Indeed, bioplastics and biodegradable systems are quite appealing but, at the moment, are far from solving the problem.

## 3. Detection of Microplastics in Wastewater Plants

Wastewater plants (WWTP) can remove part of the microplastics in waste waters, depending on the treatment units. Usually, the filters and screens used in the WWTPs have large pore volume, because the influent normally has high content of organic matter and as a result, filters or sieves clogging can happen if large volumes are collected [12].

Since, it has been shown that microplastics could pass through the WWTP, entering into the aquatic water bodies and finally accumulating in the environmental [4,46] procedures to detect their presence are crucial. The detection procedure usually involves three steps, as can be seen in Figure 6; being the methods used in each step not standardized so far [4].

In general, the analysis of microplastics can be divided into physical characterization and chemical characterization [7]. The physical characterization mainly refers to characterization of the size distribution of microplastics as well as assessing other physical parameters such as shape and colour. On the other hand, chemical characterization was applied to explore the composition of microplastics, by destructive methods (gas chromatography coupled to mass spectrometry (GC-MS), including pyrolysis-GC-MS and thermal extraction desorption-GC-MS, and liquid chromatography (LC)) or non-destructive techniques (FTIR spectroscopy, Raman spectroscopy and Scanning Electron Microscopy (SEM) [4]. Thus, the development of methods and materials able to improve detection, but also removal of these heterogeneous microparticles (granular, fragment, fiber, film, foam, etc.) are of major importance.

## 4. Removal of Microplastics: From Laboratory to Large Scale

In order to preserve the different ecosystems affected, in particular the marine environment, it is becoming increasingly important to develop different methodologies for an efficient removal of microplastics from wastewaters, which otherwise will inevitably reach the oceans through pipelines, ships, and other human activities.

In this respect, Coppock et al. (2017) developed a portable small scale method to separate microplastics from sediments of different types, using the principle of density floatation, useful for studying the abundance and properties of the separated microplastics [9]. In another work, Him et al. (2012) described the construction and application of a small-scale, portable microplastic extraction unit that mirrors the design of the Munich Plastic Sediment Separator (MPSS) [48]. The authors compare the viability and the cost of the process for three high-density salt solutions (i.e., sodium chloride, sodium iodide, and zinc chloride) to be used in the unit. The efficiency of the unit was also tested by artificially spiking sediment with known quantities of microplastics (i.e., polyethylene, polyvinyl chloride, and nylon) and validating its use with environmental samples of different types [9].

Another small scale process very useful for sampling microplastics in marine sediments, Sediment-Microplastic Isolation (SMI), was also introduced by Coppock et al. (2017) (Figure 7) [9]. Briefly, in the SMI approach, the microplastic extraction from a sediment starts by placing the equipment inside a laminar fluid cover. On each trial, a dry (30–50 g) sample, clean magnetic stir bar and a ZnCl_2_ solution are added to the purged SMI unit. After mixing the sediment, the unit is left to settle until the supernatant is clear. The supernatant is vacuum filtered and filters are transferred to clean Petri dishes to be examined by optical microscopy [9]. In order to validate the procedure, extractions of microplastic from artificially spiked sediments were also studied, the plastics being inspected for signs of degradation after extraction. Lastly, extractions were also made from local samples of natural sediments with different grain sizes.

The results demonstrated the ability of the SMI unit to extract microplastics from sediments, in a single step, with a mean recovery performance of 95.8%. The authors claimed that an optimised method can be applied for a wide range of sediment types. Moreover, it can be used in laboratory or in field to isolate microplastics from benthic samples.

Magni et al. (2019) reported the extraction and characterization of microplastics at the outlet of a waste water treatment plant in Italy [49]. After the digestion of the organic matter present in each, wastewater of sludge was collected. The samples were decanted and filtered with cellulose nitrate membranes. The method showed good removal efficiency (ca. 84%), but a large amount of microplastics was still being released into fresh water, eventually ending up as soil contaminants.

Nuelle et al. (2014) have also described a two-step method to extract microplastics (i.e., PP, PVC, and PET) from sediments, which is suggested to be suitable to monitoring microplastics in marine sediments [50]. The technique consists in two steps: (i) fluidisation of sediments in a saturated NaCl solution, and (ii) subsequent flotation of microplastics in a high-density salt solution (NaI). In addition, different solvents were also tested to dissolve biogenic matter. With this two-step approach, the NaCl pre-extraction decreases the mass of the original sediment sample, while the NaI use in the subsequent flotation of the microplastics was found to efficiently extract common synthetic macromolecules, including high density polymers. Compared with other systems based on flotation in high density salts, this method is more eco-friendly, it incurs low material costs, and the equipment is simple to obtain, making the whole process cost-effective [50]. The device for applying the AIO/flotation method (Figure 8) can be easily established in laboratories to monitor the occurrence of microplastics in sediments, even in countries with limited laboratory equipment available. Since marine plastic pollution is a global threat problem, this system introduces great advantages [51].

A slightly different method was proposed by Besley et al. (2017) who used a fully saturated salt solution of NaCl and filtration for the extraction of microplastics [52]. Filtration of the salt-solution is a key step, as previous studies found microplastic pollution in table salt [53]. Firstly, the extraction of the microplastics is achieved by density separation of the dry sand combined with saturated salt solution. The supernatant is then submitted to a vacuum filtration system and the filter membranes, containing the microplastic particles, examined by stereo-microscopes. This method allows for quantification of microplastics in the range of 0.3–5 mm [52].

A different approach was used by Ziajahromi et al. (2017) who developed a method for wastewater effluents [54]. This method consists of a high-volume sampling device with multiple mesh screens to collect a wide size range of microplastics from wastewater effluents. This process is combined with an efficient sample processing method using organic matter digestion, density separation, and staining to eliminate non-plastic particles prior to identification of microplastics in wastewater by FT-IR spectroscopy [54]. The efficiency of the device was very satisfactory as the capture of microplastics ranged from 92% for the 25 µm mesh screen to 99% for the 500 µm mesh screen. This demonstrates that this sampling device is suitable to capture a wide range of particle sizes.

Unlike the previous methods, Misra et al. (2019) developed a novel procedure using ionic liquids [55]. This approach is based on nanoparticles with a core of magnetic iron oxide and a shell of porous silicon dioxide. The surface of the nanoparticles was coated with a layer of an ionic liquid. As the used ionic liquid is in a molten state at room temperature, this method avoids the use of common solvents. The ionic liquid used was based on polyoxotungstate anions. As counterions, the bulky tetraalkylammonium cations were selected due to their antimicrobial properties. The resulting ionic liquid forms stable thin layers on the porous silicon dioxide surface of the nanoparticles. Once loaded with contaminants, including microplastics, the nanoparticles can be simply magnetically extracted from the aqueous medium, as illustrated in Figure 9.

Laboratory tests demonstrated the efficient removal of lead, nickel, copper, chromium, and cobalt ions, as well as the dye Patent Blue V, as a model for organic impurities. The growth of various bacteria was also stopped. In addition, the nanoparticles attached themselves to the surface of polystyrene spheres with diameters ranging from 1 to 10 µm, a model for microplastics, which could then be removed and quantified. Adjustment of the components of the nanoparticles should allow for further optimization of their properties, making them highly promising for both central and decentralized water purification systems. The authors conclude that this system may easily allow purification of large volumes of water, even without an extensive infrastructure.

Membrane filtration systems are also used as tertiary treatment solutions in order to remove plastics from effluents, in particular, plastic particles of smaller size, in integrated wastewater treatment systems [56]. However, polymeric membranes can themselves be a source of plastic waste. The reuse and recycling of the membranes can reduce the environmental impact and water treatment costs. Additionally, the efforts are increasingly oriented to develop membranes using new bio-based polymers (recyclable and biodegradable) as an alternative to petrochemical polymers [57].

Another method was studied by Perren et al. (2018) for the removal of microplastics from wastewater (under laboratory conditions) by electrocoagulation [58]. The effect of water characteristics, such as pH, current density and conductivity, concentration and particles size of microplastics, on the removal efficiency were thoroughly studied. The results demonstrated that at a neutral pH, the pollutant removal is improved (ca. 99%) due to the higher production of coagulant. Water conductivity showed no obvious impact on removal efficiency, and the removal efficiency increased with time and reached a steady state. The same method was also used to remove microplastics from drinking water. However, some limitations exist in this removal process, considering the removal efficiency for some operating conditions, the capacity of the treatment units and also the operating costs [59].

Chen et al. (2020) have also described an interesting flocculation method for removal of nanoplastics with different salt-based flocculants, such as aluminium and calcium. Aluminium ions have been used as highly efficient flocculants in sewage treatment while calcium ions show excellent sedimentation performance for impurities under high pH conditions. It has been demonstrated that flocculation occurring between composite metal calcium-aluminium (Ca/Al) ions and nanoplatsics showed the highest performance, particularly for high pH values [60].

## 5. Flocculation and Its Relevance on Particles Separation: Background

A typical approach considered for water purification relies on flocculation, since it is usually simple, inexpensive, and effective [61]. Different chemical agents, flocculants, are often used in fast solid–liquid separations by an aggregation process of colloidal particles [62]. These flocculants promote particle aggregation through different mechanisms, highly dependent on the flocculant type, the nature of the material to be flocculated, and the conditions of the medium. Different mechanisms can occur during flocculation (Figure 10): charge neutralization (coagulation) involves reduction of the electrostatic repulsion between particles through use of a salt or low-molecular-weight polyelectrolytes [63]. Polymers with low or medium molecular weight and high charge usually form polymer patches on an oppositely charge particle surface, resulting in flocculation through screening the electrostatic effect [64]. Bridge formation occurs when polymers with high molecular weight are adsorbed in extended conformation onto particle surfaces, forming long loops and tails, which interact and form bridges among suspended particles at concentrations below the isoelectric point [65]. Hydrogen bridge formation occurs with natural polymers, forming semicolloidal aggregates based on hydrogen bonds with polar molecules on the colloidal surface [63].

Coagulants neutralize the repulsive charges between particles and colloids, and allow bigger aggregates to be settled. The most widely used flocculants are synthetic water-soluble polymers, based on polyacrylamide and its derivatives. Polyelectrolytes with a large number of charges along the polymer chain can interact with charged particles of wastewater and destabilize the aqueous dispersion by charge neutralization, leading to the particles sedimentation and, thus, clarifying the system [66]. However, the use of synthetic flocculants can induce environmental contamination caused by residual unreacted monomers, which are typically toxic and non-biodegradable [67].

It has been show that the flocculation process can be followed by laser diffraction spectroscopy (LDS), which monitories the mean size distribution of the flocs, over time, enabling the evaluation of the kinetics of the flocculation phenomena [68]. Additionally, it allows inferring on the flocs structure, either by assessing their time resolved fractal dimension or their scattering exponent (SE) [69]. The SE parameter provides a mean of expressing the degree to which primary particles fill the space within the nominal volume occupied by an aggregate, that is, its compactness [68,70].

This strategy allows pre-screening the best flocculants for a certain application, without requiring expensive pilot tests. If we focus on contamination by microplastics of wastewater discharged from water treatments plants, flocculation can be a cheap, easy and attractive solution for microplastics removal.

Moreover, as it will be next discussed, bio-based systems capable to separate and remove pollutants, such as microplastics, from water appear as very appealing solutions.

## 6. Bio-Based Solutions for Microplastics Removal: Current Trends and Future Perspectives

In literature, the reasons identified regarding the lack of bio-based solutions in the microplastic flocculation and removal are mainly related with the poor water solubility and low charge density that such bio-based systems often present [71]. These less suitable physicochemical features typically result in systems of poor efficiency. Therefore, in order to extend the applicability of bio-based systems and maximize their performance, it is essential to start rationalizing strategies aiming at the water solubility and charge density improvement of those natural additives. In what follows, different approaches are suggested to address the identified issues.

Lignin and cellulose, due to their favourable properties, such as vast abundance, inexpensiveness and renewability, are obvious candidates as bio-based solutions to tackle the microplastic threat. The introduction of charged groups into lignin or cellulose has been shown to increase their water solubility, enhancing their interactions with species of interest [67,72]. For cationization of natural polymers, amination is one of the possible strategies that can improve the water solubility and charge density of natural polymers [73]. In cellulose, the reaction occurs on the hydroxyl groups of the beta-glucan units (Figure 11), thus allowing the potential grafting of new functional groups and extending the range of practical applications [74].

A different modification method was recently suggested by Grenda et al. (2019) to introduce cationic charges along the cellulose chain [75,76,77]. The authors used a two-step reaction to produce cationic cellulose, based on a periodate oxidation followed by the reaction between the Girard’s reagent and the aldehyde groups of the oxidised cellulose [77] (Figure 12). The resulting modified polymers, with different cationic charge densities, were successfully employed as flocculants and coagulants of dyes [76,77] and silica nanoparticles [75]. The possibility of using their type of modified cellulose in microplastic flocculation and removal is thus highly promising.

The insertion of cationic groups into lignin has been shown to increase its solubility in water [72]. Ronny et al. (2017) demonstrated that the amination of lignin with dimethylamine produces a highly cationic lignin derivative. The charge and coagulation efficiency of the aminated lignin are directly dependent on pH, as low pH leads to protonation of the amine groups. Cationization with glycidyltrimethylammonium chloride (GTAC) introduces quaternary ammonium groups into the lignin backbone, which are charged irrespective of the pH [67] (Figure 13). The resulting modified lignin molecules have been reported to be useful flocculants for dye removal in waste waters [72].

Grenda et al. (2018) also obtained tannin-based coagulants capable of removing cationic and anionic dyes from aqueous media using a two-step procedure [78]. In the first step, a reactive iminium ion is formed by the acid-catalysed addition of dimethylamine hydrochloride to formaldehyde, while on the second step the iminium ion is supposed to react with the enol groups of the tannins, forming modified tannin products. The novel tannin-based derivatives were successfully tested regarding the potential for colour removal from aqueous solutions containing cationic (i.e., methylene blue and crystal violet) and anionic (i.e., acid black 2 and duasyn direct red) dyes. In principle, this procedure is suitable to be extended to other polyphenols, such as lignin.

Wei et al. (2008) used other natural polymer, starch, to develop new cationized flocculants suitable for wastewater treatment [62,79] (Figure 14). The best performance in the flocculation of aqueous kaolin suspensions was found for modified starch with longer aliphatic side chains, which was even superior to a commercially available flocculant agent. The results of Wei et al. also suggest that the hydrophobicity of the coagulant agent can play an important role in separation processes [62].

A possible route to change the hydrophobicity of polymers containing hydroxyl groups is based on its acetylation. In this respect, Jonoobi et al. (2010) performed the acetylation of the hydroxyl groups of cellulose nanofibres to reduce its hydrophilicity [80] (Figure 15).

The acetylation of cellulose depends on the accessibility and susceptibility of the OH-groups in the amorphous and crystalline regions within the cellulose fibrils. It is important to note that the produced cellulose nanofibres are not water soluble and thus their application as flocculants in aqueous medium is not straightforward. However, if the acetylation degree is adequately controlled so that the solubility of the cellulose fibres is not compromised, this approach can represent an interesting bio-based solution to remove microplastics from aqueous media, particularly if the cellulose nanofibres are processed in the form of membranes or aerogels [80].

Besides cellulose and starch, chitin is another polysaccharide that is considered promising as a reliable future bio-based flocculant [81]. For example, Chen et al. (2010) used 3-chloro-2-hydroxypropyltrimethylammonium chloride to introduce cationic charges on chitin. The modified polymer became water-soluble, thus now being suitable to be used as flocculant in water treatment [79].

The studies described above demonstrate the potential of different biopolymers as flocculation agents. However, none of the reported studies were specifically directed to the flocculation of microplastics as a strategy to remove microplastics flow water systems. Using a cationic hydrophobically modified cellulose derivative as bio-flocculant, the authors have performed preliminary tests of the flocculation of micronized plastic in aqueous medium. To the best of our knowledge, these are the first evidences of the flocculation of microplastics using bio-based flocculants. The model system was micronized polyethylene terephthalate (PET) produced in our lab, with a slight inherent negative surface charge, and an average particle size of ca. 212 m. We applied laser diffraction spectroscopy (LDS) to monitor the flocculation process in slight turbulent conditions following previous developments from the group (simulated by stirrer). As referred, this technique can supply information about the flocculation kinetics [70]. The tests were conducted in a Malvern Masterziser 2000 (Malvern Instruments). The “contaminated” water sample was produced in our lab by addition of 0.1wt% microplastics to 700 mL of distilled water. The measurements (size distribution) of the initial effluent were carried out at a stirring speed of 1000 rpm. The flocculant was added after 6 min from the beginning of the experiment, with dosages (w/w) of 0.1% and 0.01% based on the concentration of microplastics.

The pre-determined amount of flocculant solution was added at once to the “contaminated sample” using a stirring speed was decreased to 500 rpm (15min after starting the experiment). Due to the reduction of the stirring velocity, the average size increases considerably to ca. 450 µm. Note that the lower stirring speed allows flocs to come together more efficiently, forming bigger aggregates. The same qualitative behaviour was found for other concentrations of flocculant agent. Different stirring speeds were also tested, and 500 rpm was found to have the best compromise between the formation of the largest flocs, while ensuring their circulation in the system without sedimentation. Figure 16 presents the evolution with time, of particle size in the system, obtained through LDS, for two concentrations of the bio flocculant.

From Figure 16, it becomes evident that the used bio-based flocculant can efficiently interact with PET microparticles and form large flocs. These flocs can be easily removed by sedimentation or using a standard filtration method. Also it was demonstrated that the concentration of flocculant is crucial to obtain a good flocculation. Excess of flocculant can be detrimental to the removal process, making flocculant adsorption on the particles surface more difficult as has been described in the literature, and thus retarding the flocculation process. This novel strategy enabled not only an efficient aggregation and removal of microplastics, but also the precise monitoring of the kinetics of flocculation and size variation of aggregates.

From this brief overview, it is clear that the biopolymers abundantly available in nature may constitute a sustainable and efficient solution to address the eminent microplastic threat. According to the studies described above, the mechanism of flocculation/coagulation involves electrostatic interactions between oppositely charged species. However, it is important to keep in mind that microplastics can be neutral species or materials of low charge density, which may reduce the efficiency of the separation process. Apart from the electrostatic interactions, hydrophobic effects are also expected to contribute to the aggregation phenomenon in aqueous media. Therefore, in order to obtain materials with superior flocculation properties, the development of amphiphilic-like polyelectrolytes appears as a reasonable solution. Adopting state-of-the-art strategies to equip these biopolymers with suitable functional groups may enable their massive use as superior flocculants for microplastics removal.

## 7. Concluding Remarks

Even without knowing, people are generally exposed to high levels of pollution by microplastics, and thus, this does mean that societies cannot relax. The number of uncertainties and research gaps is much higher than the current knowledge to simply ignore this “invisible” reality.

Nowadays, the wastewater and drinking water treatment systems are pretty much the only approach to minimize microplastic dissemination and, according to the available data, these systems are clearly deficient. It is evident that microplastics are present in all water circuits but their fate is uncertain because some of them are defragmented by increasing water contamination and others are ingested by marine biota possibly resulting in adverse health effects.

Bioplastics can serve as a biodegradable and less carbon-intensive sustainable alternative for traditional plastics, while also making use of materials, such as food scraps that might, otherwise, be discarded. Their use remains minimal, however, mainly due to the lack of efficient new technologies required and high cost of production. This raises the question of how viable the bioplastics are in the near future. Moreover, some experts claim that bioplastics may not be environmentally preferable after all, focusing on the fact that some forms do not biodegrade more easily than conventional plastics, or because their production requires the use of land and resources that could otherwise be used to grow food for people. For these reasons, we see an important and necessary scientific interest growing in the search for efficient methodologies that can extract/remove/reduce the microplastics contaminating our planet, mainly the watercourses and oceans. In this respect, biopolymer-based flocculant solutions can, in the future, represent a reliable alternative to help tackle the microplastics potential threat.

## Figures and Tables

**Figure 1 molecules-25-03954-f001:**
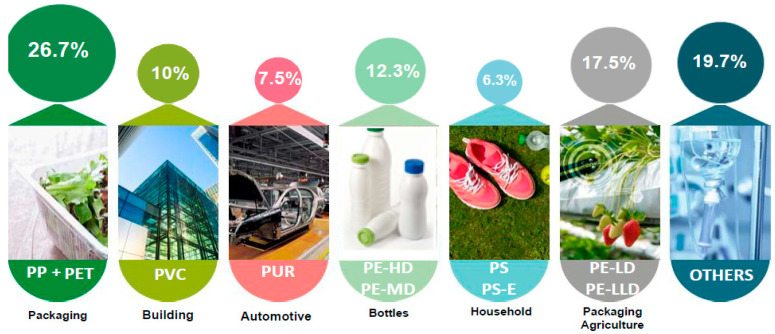
Worldwide plastic demand by polymer type. Adapted from reference [3].

**Figure 2 molecules-25-03954-f002:**
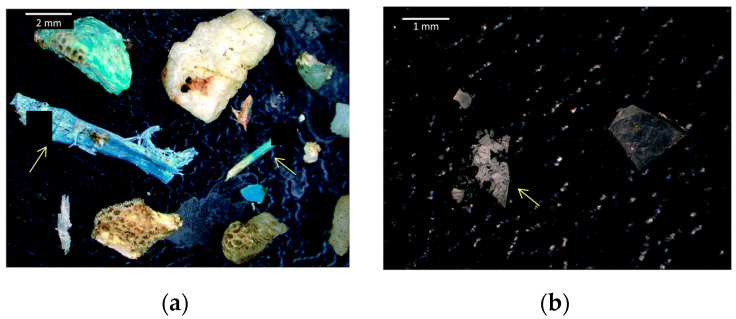
Images of Pacific Ocean trawl microplastic particles, **a**) primary and **b**) secondary microplastics, some with adhered crustaceans, mineral crusts, or radiolarians (taken from reference [19] with permission of the Royal Society of Chemistry).

**Figure 3 molecules-25-03954-f003:**
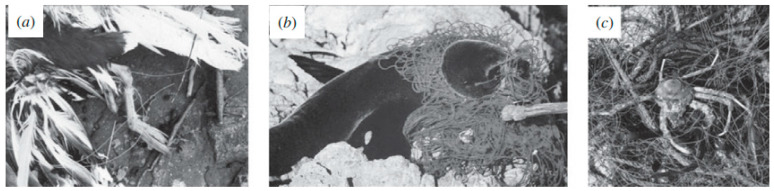
Examples of animals that ingested microplastics: (**a**) southern black-backed gull, Larus dominicanus caught and hooked in nylon filament fishing line; (**b**) a New Zealand fur seal trapped in discarded netting and (**c**) Ghost fishing-derelict fishing gear dredged from >100 m on the Otago shelf (taken from reference [23] with permission of the Royal Society).

**Figure 4 molecules-25-03954-f004:**
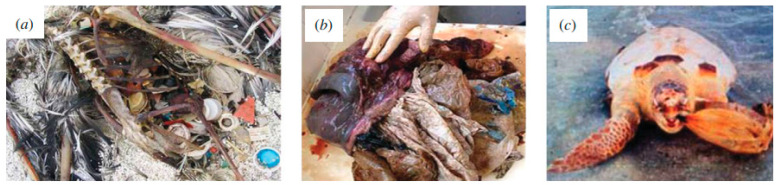
Examples of ingestion: (**a**) Laysan Albatross; (**b**) plastic from the stomach of a young Minke whale that had been washed ashore dead in France and (**c**) stranded sea turtle disgorging an inflated plastic bag. (taken from reference [23] with permission of the Royal Society).

**Figure 5 molecules-25-03954-f005:**
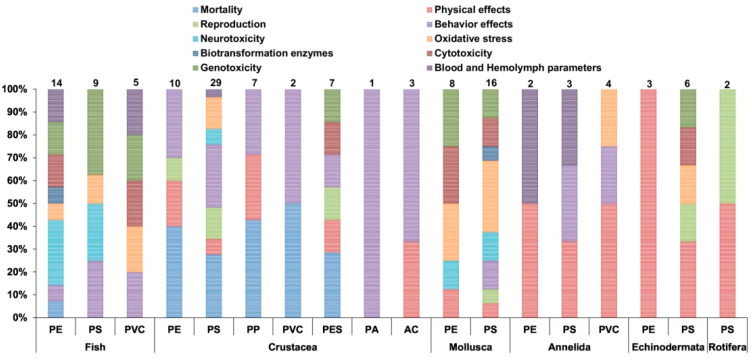
Ecotoxicological effects of microplastics on the different groups of organisms. Each bar has the total number of studies on it. [32].

**Figure 6 molecules-25-03954-f006:**
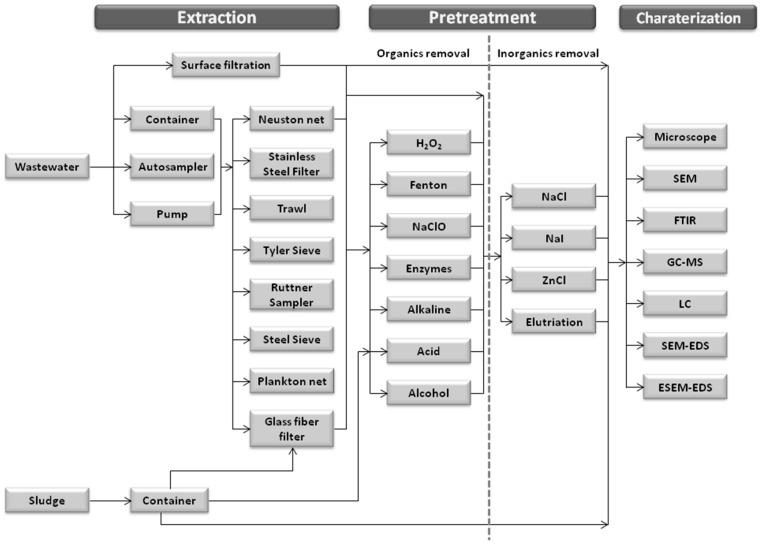
Flow chart summarizing steps and techniques used for microplastics detection in WWTPs (adapted from reference [4,47]).

**Figure 7 molecules-25-03954-f007:**
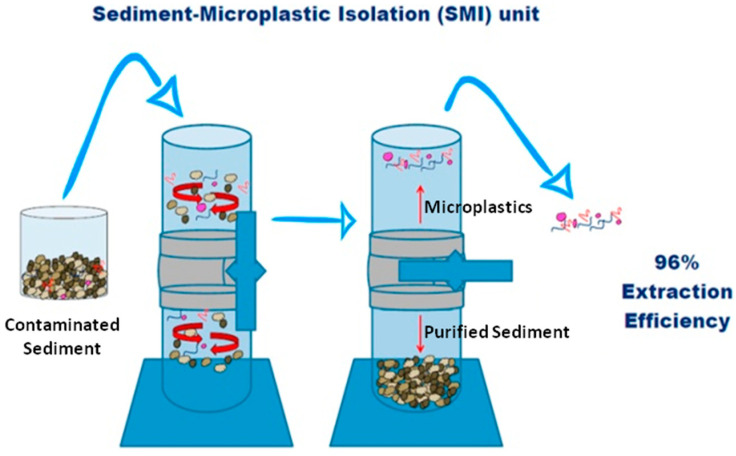
Schematic representation of the SMI approach (taken from reference [9] with permission from Elsevier).

**Figure 8 molecules-25-03954-f008:**
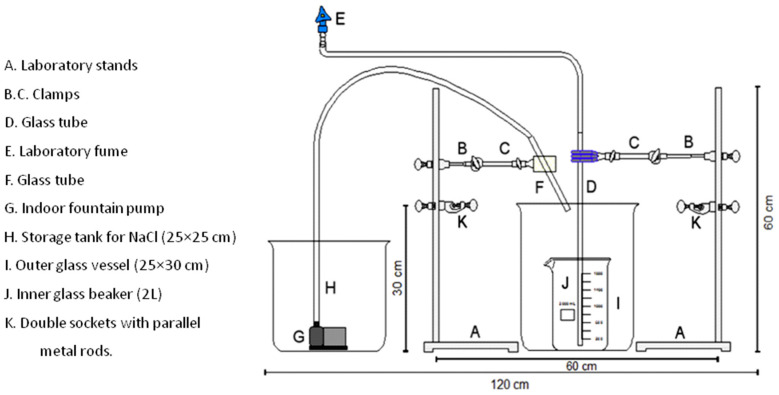
Scheme of the experimental setup for the air-induced overflow AIO method (taken from reference [50] with permission of Elsevier).

**Figure 9 molecules-25-03954-f009:**
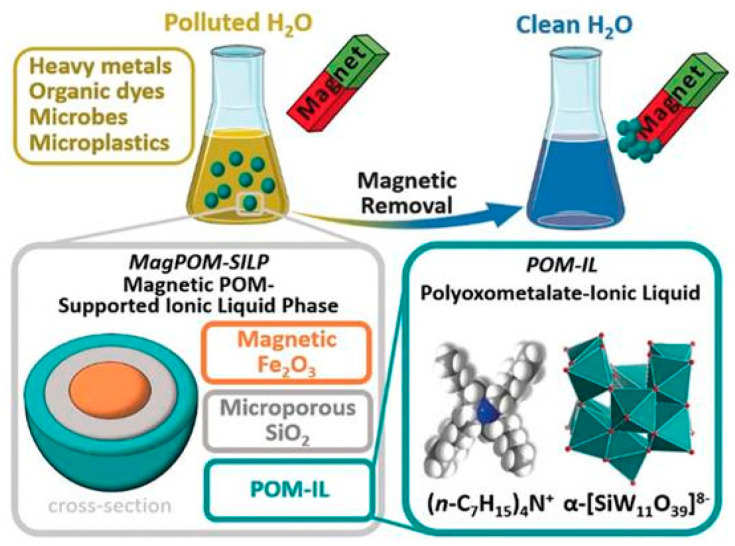
Schematic representation of multiple pollutants removal from water using magnetic polyoxometalate supported ionic liquid phases (taken from reference [55] with permission of John Wiley & Sons, Inc).

**Figure 10 molecules-25-03954-f010:**
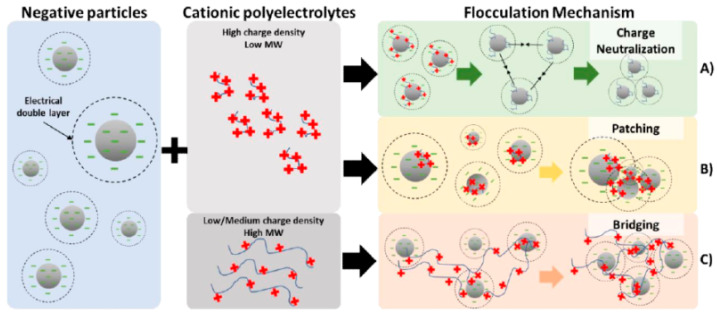
Illustration of the hypothetic flocculation mechanisms occurring among negatively charged particles and cationic polyeletrolytes: (**A**) charge neutralization, (**B**) patching and (**C**) bridging.

**Figure 11 molecules-25-03954-f011:**
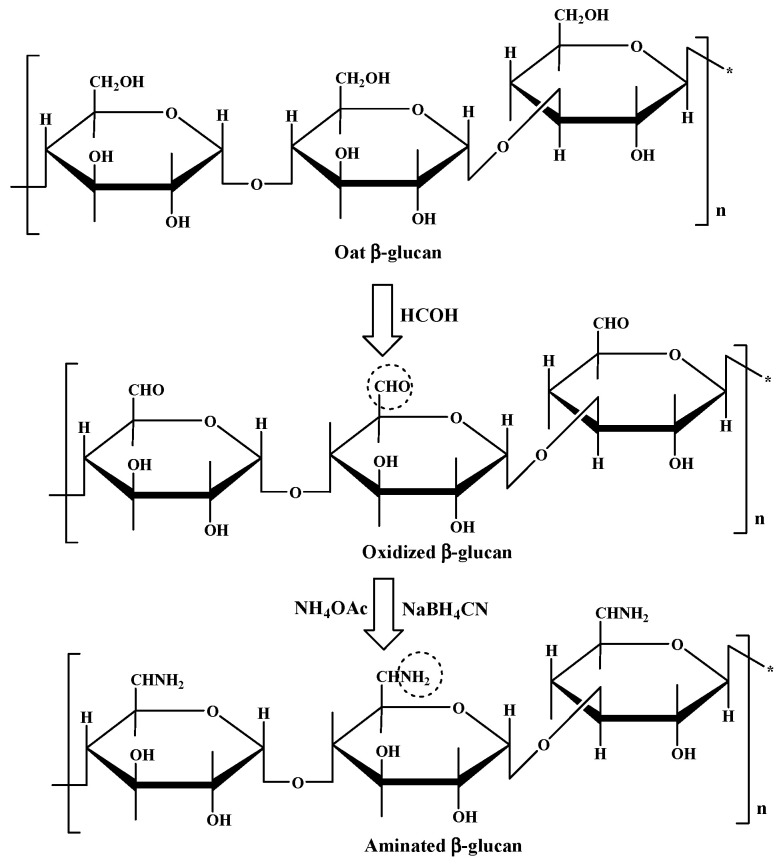
Schematic representation of reductive amination of b-glucans. Taken from reference [74] with permission of the American Chemical Society.

**Figure 12 molecules-25-03954-f012:**
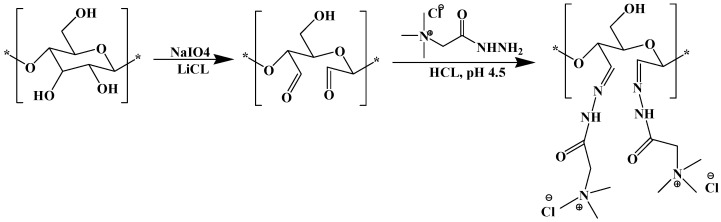
Two-step reaction scheme used to produce cationic cellulose (Taken from reference [77] with permission of the Royal Society of Chemistry).

**Figure 13 molecules-25-03954-f013:**
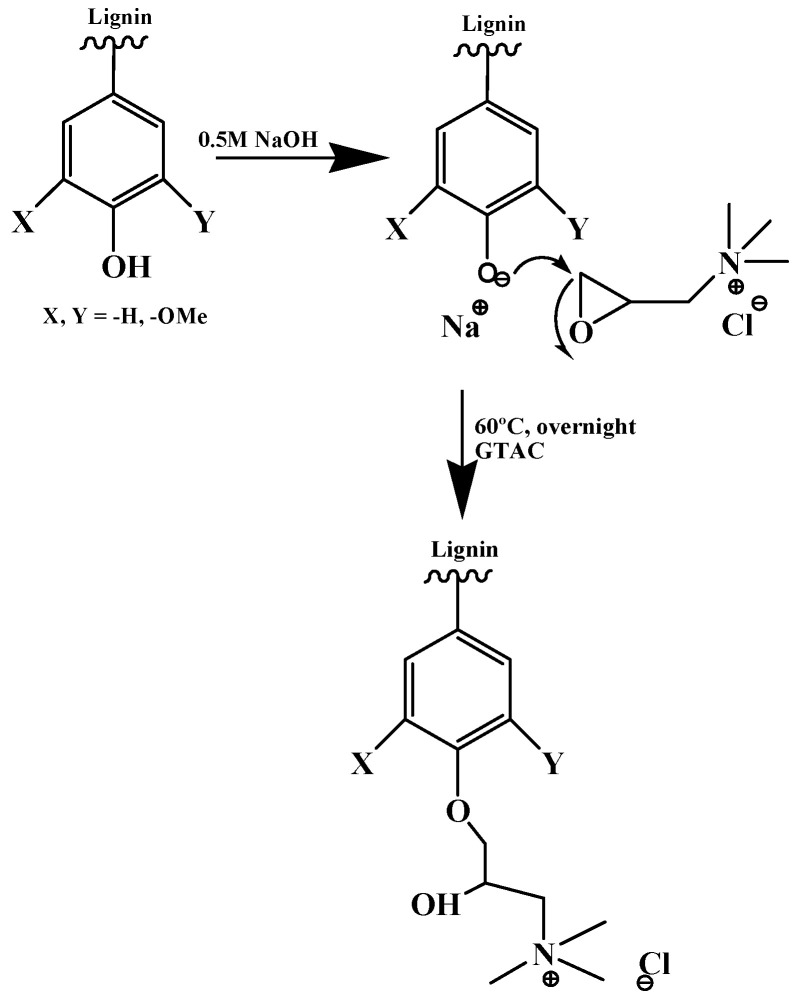
Reaction scheme for lignin derivatization with GTAC under alkaline conditions (Taken from reference [67] with permission of the European Polymer Journal).

**Figure 14 molecules-25-03954-f014:**
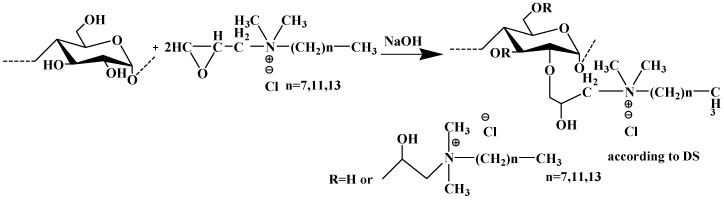
Synthesis scheme of cationic starch derivatives. Taken from reference [62] with permission of Elsevier.

**Figure 15 molecules-25-03954-f015:**
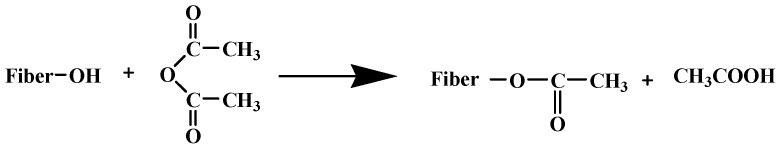
Chemical reaction of acetic anhydride with cellulose (Taken from reference [80] with permission of the Springer Nature).

**Figure 16 molecules-25-03954-f016:**
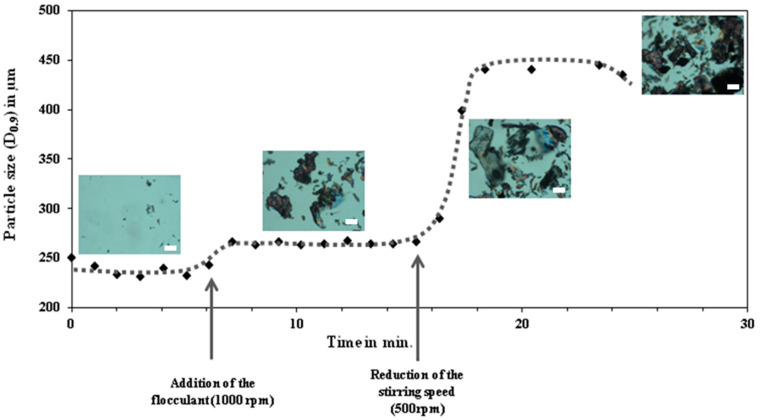
Evolution of particle size (D_0.9_) as function of time with different concentrations of flocculant (*w*/*w*) 0.01% (top) and 0.1% (bottom) based on the concentration of microplastics. The scale bars in the inserts represent 100 µm.

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
