# Peer review of "Microplastics in Ecosystems: From Current Trends to Bio-Based Removal Strategies"

_molecules, 2020, doi:10.3390/molecules25173954_

Round 1
Reviewer 1 Report
The authors summarized the bio-based solutions for removal of microplastics. The reviewer would like to request to revise several figures because their text fonts are not easy to read. Please revise text in Figures 5,7,8,9,10,12,15.
Author Response
We are thankful to the reviewer for recognizing the importance of our manuscript. We agree with the reviewer suggestion and the indicated figures have been revised. Nevertheless, we believe that most of the issues related to the low quality of the text fonts is due to the word-to-pdf file conversion. The original images have been uploaded to guarantee the best quality possible.

Reviewer 2 Report
This review is summarized the method for the removal of microplastics in eco-systems. However, the amount of intelligence is poor. I do not recommend the acceptance of the manuscript in “Molecules”.
1) I think that the title is not suitable for the contents of the manuscript. The information about the removal of microplastic in eco-systems is poor.
2) Most of the references in the manuscript is not new and updated. This area has been actively studied, and many reports has been published. The authors should cite many recent reports.
e.g.
Chemosphere 252 (2020) 126450
Proc R. Soc. B. 285, 2018 1203.
etc.
3) The title of chapter 2 is ‘Microplastics and their impact”. However, the chapter mainly describes the impact of plastic debris, not microplastics. Many reports described the toxicity of microplastic on organisms have been published. The authors should cite the reports about the toxicity of microplastics and discuss the impact of microplastics on eco-systems.
4) Chapter 3, 4, and 5 did not describe the bio-based solutions for removal of microplastics in ecosystems.
Author Response
1) I think that the title is not suitable for the contents of the manuscript. The information about the removal of microplastic in eco-systems is poor.
Reply: Following the reviewer comment, the title has been changed to better fit the paper content. Nevertheless, we highlight that our manuscript summarizes the most relevant works dealing with microplastics removal, from lab scale to the efforts made on implementing technologies in large scale, even if information on large scale removal is very scarce in the literature. Although the majority of the existing literature cited in our manuscript is related with lab scale approaches, this was also an important driving force to highlight the few developed bio-based systems solutions which are promising solutions to be applied in large scale wastewater plants and minimize the microplastics threat.
2) Most of the references in the manuscript is not new and updated. This area has been actively studied, and many reports has been published. The authors should cite many recent reports.
e.g. Chemosphere 252 (2020) 126450
Proc R. Soc. B. 285, 2018 1203. etc.
Reply: The manuscript has been updated with more recent relevant references, such as the one suggested by the reviewer concerns, even if there were already many recent references, 2015 onwards, in the initial review. Changes are properly highlighted in the revised text.
3) The title of chapter 2 is ‘Microplastics and their impact”. However, the chapter mainly describes the impact of plastic debris, not microplastics. Many reports described the toxicity of microplastic on organisms have been published. The authors should cite the reports about the toxicity of microplastics and discuss the impact of microplastics on eco-systems.
Reply: We thank the reviewer comment but we do not fully agree with it. It is true that, this section also discusses the health problems caused by plastic debris and therefore the title has been changed to meet the reviewer concerns and better fit the section content. Nevertheless, we stress that the main focus of this section is on the effects of microplastics in ecosystems, marine and terrestrial systems, and even humans. Some striking examples are here recalled to highlight our position“...it has been observed that the absorption of microplastics and nanoplastics by humans can lead to a wide range of organism’s obstruction, inflammation and accumulation in organs after translocation”, “...reduction in photosynthesis of plants and effects on the feeding activity of zooplankton and marine animals (adverse effects to gill, stomach and alterations in histology) have also been argued to occur after microplastic intake”.
4) Chapter 3, 4, and 5 did not describe the bio-based solutions for removal of microplastics in ecosystems.”
Reply: The reviewer is right but we believe these chapters are fundamental not only to contextualize the problem but also to provide useful information to understand microplastics and how to deal with this problem. While chapters 3 and 4 discuss the current detection and extraction methodologies, chapter 5 highlights the intermolecular interactions and mechanisms that govern the flocculation process, essential in many particle separation processes. By doing so, we provide a robust scientific support and background to chapter 6 in which we introduce several potential applications of bio-based systems with potential application for microplastics removal. These possible solutions are introduced together with our proof-of-concept where a cellulose derivative is successfully used to flocculate plastic microparticles.

Reviewer 3 Report
The paper addresses a topic of great importance today: the presence of microplastics in surface waters due to excessive pollution with products of this kind. The authors review the literature and briefly present studies conducted in recent years. It also presents as a method of reducing and even removing microplastics from water, the use of bio-polymers. The paper is well documented and is interesting for specialists in the field but it is accessible to non-specialists as all notions are very clearly explained. However, I think that for such an important scientific journal the approach is a bit simplistic. I think the article should be enriched with more complex studies.
Author Response
We are thankful to the reviewer for the very positive feedback and recognizing the general importance of our manuscript. Indeed, it was our intention to keep the paper as simple as possible without compromising the scientific content and providing as much information as possible on the existing approaches, performing a thorough literature review. This approach was chosen not because the microplastics and related issues are simple problems but rather to reach a vaster audience even not familiar with some of the key concepts discussed. By doing so, we believe this can bring more attention to the microplastics problem and trigger novel R&D strategies to face their potential threat to ecosystems. Although we understand the reviewer point, we have deliberately avoided the discussion of “complex studies”, even if we do not fully understand what the reviewer means by this, because we are strongly convinced that in order to minimize the microplastic’s threat worldwide, any affordable strategy has to be as simple as possible. Complex approaches typically involve high costs of production, challenges in scale-up, impossibility to implement on a global scale, etc. We believe the systems discussed are generally simple and some of them highlight promising, viable solutions to handle the microplastic threat. Still, some new references were added to the manuscript.

Round 2
Reviewer 2 Report
I think that the authors do not answer my suggestion.
In particular, the toxicity of microplastics should be added in the manuscript instead of the toxicity of plastic debris.
Author Response
We apologize not following earlier the reviewer suggestion. We now believe the manuscript provides an improved discussion on the effects of microplastics on different ecosystems and human health.

Reviewer 3 Report
I think you are right about the simplistic way of approaching this subject, if it is addressed to a wider audience. My objection started from the fact that Molecules journal is addressed especially to specialists and very little to the general public.
However, if the editors of the journal consider that this approach is appropriate then they agree with the publication of the paper.
I agree with the publication of the paper.
Author Response
We deeply appreciate the reviewer comment.
